# Mapping mutational fitness effects across the coxsackievirus B3 proteome reveals distinct profiles of mutation tolerability

**Beatriz Álvarez-Rodríguez, Sebastian Velandia-Álvarez, Christina Toft, Ron Geller** *

Institute for Integrative Systems Biology (I2SysBio), Universitat de Valencia-CSIC, Valencia, Spain

* ron.geller@csic.es

**Citation:** Álvarez-Rodríguez B, Velandia-Álvarez S, Toft C, Geller R (2024) Mapping mutational fitness effects across the coxsackievirus B3 proteome reveals distinct profiles of mutation tolerability. PLoS Biol 22(7): e3002709. https://doi.org/10.1371/journal.pbio.3002709

**Data Availability Statement:** All scripts and data are available on the project's GitHub (https://github.com/RGellerLab/CVB3_full_proteome_DMS; doi:10.5281/zenodo.11550038). Sequencing data

## Abstract

RNA viruses have notoriously high mutation rates due to error-prone replication by their RNA polymerase. However, natural selection concentrates variability in a few key viral proteins. To test whether this stems from different mutation tolerance profiles among viral proteins, we measured the effect of >40,000 non-synonymous mutations across the full proteome of coxsackievirus B3 as well as >97% of all possible codon deletions in the nonstructural proteins. We find significant variation in mutational tolerance within and between individual viral proteins, which correlated with both general and protein-specific structural and functional attributes. Furthermore, mutational fitness effects remained stable across cell lines, suggesting selection pressures are mostly conserved across environments. In addition to providing a rich dataset for understanding virus biology and evolution, our results illustrate that incorporation of mutational tolerance data into druggable pocket discovery can aid in selecting targets with high barriers to drug resistance.

## Introduction

RNA viruses must execute a conserved and coordinated set of functions to successfully propagate, including the delivery, replication, and packaging of genomic material in host cells as well as the modification of the host environment to favor viral spread. To accomplish these tasks, RNA viruses encode a minimal repertoire of functionally conserved proteins, among which are structural proteins involved in the formation of the viral particle (e.g., capsid or envelope proteins) and an RNA-dependent RNA polymerase required to replicate the genome [1]. In addition, most harbor additional nonstructural proteins that are involved in various aspects of genome replication (e.g., helicases) and modification of the host environment (e.g., viral proteases, viroporins).

RNA viruses have a remarkable ability to rapidly evolve and adapt to changing environments. This is mostly fueled by extreme mutation rates stemming from replication by error-prone polymerases that lack proofreading functions [2–4]. During replication, these polymerases frequently misincorporate nucleotides, resulting in amino acid (AA) substitutions, while insertions and deletions can also be introduced by various mechanisms [5]. Experimental

for this project has been deposited in SRA (Bioproject IDs: PRJNA643896, PRJNA779606, PRJNA1013170, PRJNA1033421, see GitHub section B2 for sample details). The data underlying the figures can be found in the S1 Data file.

**Funding:** The project was funded by grants PID2021-125063NB-I00 and CNS2022-135100 to R.G. from the Spanish Ministerio de Ciencia, Innovación y Universidades, the Agencia Estatal de Investigación, and the European Union (MICIU/AEI/ 10.13039/501100011033 and FEDER, UE or NextGenerationEU/PRTR). B.A-R acknowledges postdoctoral funding JDC2022-050122-I from the Spanish Ministerio de Ciencia, Innovación y Universidades, the Agencia Estatal de Investigación, and the European Union NextGenerationEU/PRTR) and the Generalitat Valenciana (APOSTD/2021/017, 2021-2023). S.V-A is funded by a Grisolia doctoral fellowship from the Generalitat Valenciana (CIGRIS/2021/080). The funders had no role in study design, data collection and analysis, decision to publish, or preparation of the manuscript.

**Competing interests:** The authors have declared that no competing interests exist.

**Abbreviations:** AA, amino acid; CAR, coxsackievirus-adenovirus receptor; CRE, *cis*-regulatory element; CTD, C-terminal domain; CVB3, coxsackievirus B3; DAF, decay-accelerating factor; dMFE, deletion mutational fitness effect; DMS, deep mutational scanning; ER, endoplasmic reticulum; ERGIC, endoplasmic reticulum-Golgi intermediate compartment; MBD, membrane-binding domain; MFE, mutational fitness effect; MOI, multiplicity of infection; NGS, next-generation sequencing; NTD, N-terminal domain; PFU, plaque-forming unit; RdRP, RNA-dependent-RNA polymerase; RPE, retinal pigment epithelial; RSA, relative surface area.

assessment of viral polymerase error rates in the absence of selection for fitness has shown that mutation rates remain high across the genome [6]. However, in nature, where selection purges deleterious mutations, mutational diversity is not uniformly distributed across different viral protein classes. Specifically, proteins targeted by adaptive immune responses, such as viral capsids or envelope proteins, often exhibit greater sequence diversity than other protein classes [7–9]. While previous studies have examined differences in the overall distribution of mutational fitness effects (MFEs) between structural and nonstructural proteins [10], a detailed analysis of how different viral protein classes accommodate mutations is lacking. Furthermore, comprehensive evaluation of how deletions affect viral protein function has only recently begun to be addressed and remains limited [5,11–13].

Picornaviruses are a ubiquitous family of positive-strand RNA viruses that infect both humans and animals, inflicting significant economic and health burdens [14]. Among these, coxsackievirus B3 (CVB3) is a human pathogen belonging to the Enterovirus B species that mostly results in mild disease but can cause severe morbidity in some cases [15,16]. Within its short genome, CVB3 encodes an array of structurally and functionally diverse proteins, including 4 structural proteins (VP1-4) that form the viral capsid and 7 nonstructural proteins (2A-2C and 3A-3D) that play diverse and essential roles in viral replication and host modulation. The structure and function of most CVB3 proteins have been extensively studied, with high-resolution structures available for most of the CVB3 proteome or related enterovirus proteins. These aspects, combined with its efficient reverse genetics system, make CVB3 an excellent choice for experimentally addressing whether mutation tolerance varies across different protein classes. Importantly, many CVB3 proteins show structural conservation across the greater picornavirus family and even other viral families (e.g., proteases or polymerases), facilitating the extrapolation of results to additional pathogens.

Deep mutational scanning (DMS) approaches provide comprehensive insights into MFE by systematically introducing a large fraction of all single AA mutations into a gene of interest and measuring their effects on fitness [17]. Previously, we applied DMS to the CVB3 capsid region (encoded in the viral P1 precursor protein) to define how mutations affected viral fitness and their correlation with structural, functional, and evolutionary parameters [18]. Herein, we extend our DMS analysis to the nonstructural proteins of CVB3 (encoded in the viral P2 and P3 precursor proteins; approximately 60% of the proteome). Together, we assess the effect of >40,000 AA mutations across the full viral proteome as well as >97% of codon deletions across the nonstructural proteins in 2 cell lines. We find significant variation in mutation tolerance within and between different viral proteins that correlate with both general and protein-specific structural and functional attributes. Interestingly, trends in MFE were similar in the 2 cell lines analyzed, revealing conservation in selection pressures across different environments. Finally, we show that MFE can be integrated into drug discovery to prioritize the selection of druggable pockets with high barriers to the development of drug resistance.

## Results

### Deep mutational scanning of the CVB3 proteome

Previously, we used a PCR-based mutagenesis protocol to introduce nearly all possible AA mutations across the capsid of CVB3 and analyzed their effects on viral fitness [18] (Fig 1A). To understand how different protein classes accommodate mutations, we extended our analysis to the nonstructural proteins. For this, DMS was performed on the P2 and P3 regions of the proteome independently, with short overlaps in the downstream (P1, 5 AA) and upstream (P3, 20 AA) regions included in the P2 mutagenesis region to enable standardization across all 3

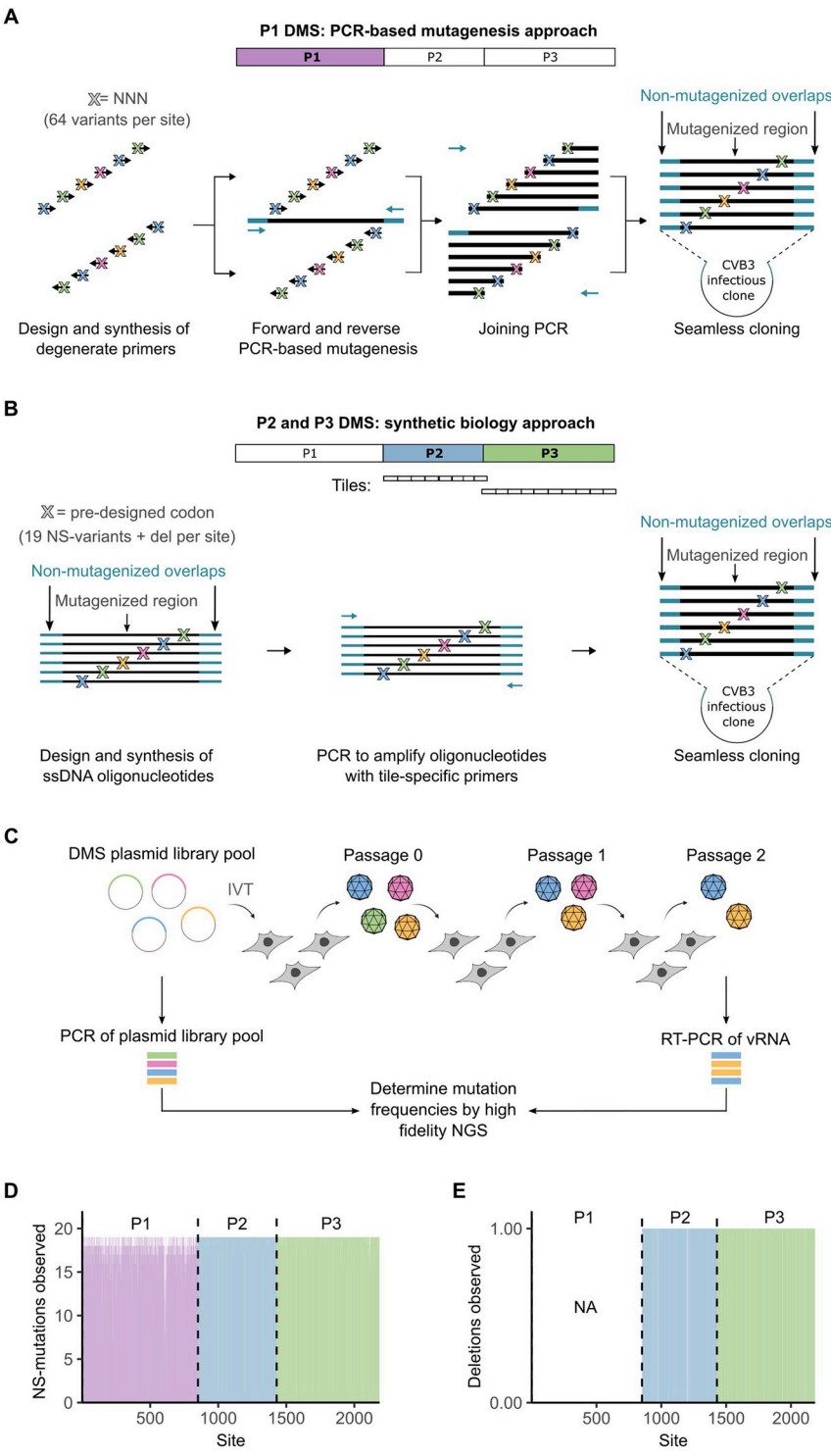

**Fig 1. Overview of the full-proteome DMS analysis. (A)** Overview of the mutagenesis protocol used for the structural proteins. A codon-level PCR mutagenesis using degenerate primers with randomized bases at the codon-matching positions (NNN) was used to mutagenize the complete P1 region, as previously reported [18]. **(B)** Overview of the mutagenesis protocol used for the nonstructural proteins. The P2 and P3 regions were mutagenized separately via a synthetic biology approach. Each region was divided into approximately 250 bp tiles. For each residue to mutagenize in the tile, a pool of oligonucleotides was synthesized encoding either a deletion or a single mutant codon for each of the possible 19 AA mutations. Stop codons and silent mutations were introduced as controls at a low frequency. All oligonucleotides harbored complementary sequences to the flanking regions in the infectious clone, which were used

to both amplify the tiles from the pool and clone them into the corresponding region in the infectious clone. **(C)** Generation of viral populations and determination of MFE by NGS. vRNA was produced from the mutagenized plasmid of each region by IVT and electroporated into HeLa-H1 cells. Following a single cycle, cells were freeze-thawed to liberate viral particles. Two additional rounds of infection were performed at low multiplicity of infection to reduce complementation and enable selection. The mutagenized region was then amplified from both the plasmid libraries as well as from vRNA from passage 2 following reverse transcription, and mutation frequencies were obtained via high-fidelity NGS. **(D)** The number of AA mutations observed for each residue. **(E)** The number of single codon deletions observed for each residue across the nonstructural region (P2 and P3). Note, deletions were not included in the mutagenesis of the P1 region. NA, not available. See GitHub [19] section A2 for sequences of synthetic oligonucleotides, S1 Table for primers used to amplify the infectious clone vector and the tiles, and S2 Table for NGS statistics. AA, amino acid; DMS, deep mutational scanning; IVT, in vitro transcription; MFE, mutational fitness effect; NGS, next-generation sequencing; vRNA, viral RNA.

regions. Rather than using the previously employed PCR-based mutagenesis method, a new synthetic biology approach was utilized for the mutagenesis, where approximately 250-bp long oligonucleotides encoding either the deletion of a given codon or a single mutant codon for each of the 19 possible non-synonymous mutations were introduced into the infectious clone (Fig 1B). In addition, stop codons and synonymous codons were introduced as controls at low frequencies. As compared to PCR-based mutagenesis, this method should significantly reduce redundancy ($n = 19$ versus $n = 64$ mutant codons with random mutagenesis), minimize background from non-mutagenized WT sequence, and limit the occurrence of multiple mutations per clone (see S1 Fig). Importantly, it enables the selection of mutant codons that have the greatest number of base changes away from the original codon, minimizing background from sequencing errors that are dominated by single substitutions [18].

Three independent mutagenized libraries were generated for the P2 and P3 regions. These were used to produce viral RNA by in vitro transcription, which was electroporated into HeLa-H1 cells to generate viral populations. Two subsequent rounds of infection (passages) were then performed at a low multiplicity of infection to reduce complementation and enable selection for fitness (Fig 1C). For the P1 region, previously characterized passage 1 viral populations [18] were used to similarly infect cells for a second round. Mutation frequencies were then obtained from both the mutagenized libraries and the passage 2 viral population using a high-fidelity next-generation sequencing (NGS) technique [18]. Mutagenesis was uniformly distributed across the proteome for AA mutations (Fig 1D) and the nonstructural proteins for deletions (Fig 1E). This enabled us to derive the mutational fitness effects (MFE) of 98% of all AA mutations ($n = 42,630$) and single codon deletions ($n = 1,305$) across the full proteome or the nonstructural proteins, respectively. Sequencing statistics are available in the supplementary S2 Table.

To calculate MFE, the relative frequency of each AA mutation versus the WT AA at each site in the viral populations was divided by its relative frequency in the plasmid libraries and the log2 values of these ratios were averaged across the independent replicates (Fig 2A; see S2 Fig for correlations between replicates). Examination of MFE for mutations present in the overlap regions between P1/P2 ($n = 61$) and P2/P3 ($n = 161$) showed a good linear relationship (Pearson correlation coefficient = 0.74 and 0.83 for P1/P2 and P2/P3, respectively; Fig 2A and S3 Fig). This allowed us to normalize MFE across the 3 regions based on a linear model. Finally, to validate our results and calibrate selection in our experiments to a more realistic measure of fitness, we generated 31 individual mutants across the full proteome and experimentally determined their fitness by direct competition with the WT virus. DMS-derived MFE correlated well with experimental measurements of fitness (Pearson correlation coefficient = 0.78, $p < 10^{-7}$; Fig 2A), enabling us to standardize our DMS results to experimentally determined fitness values using a linear model (see Methods).

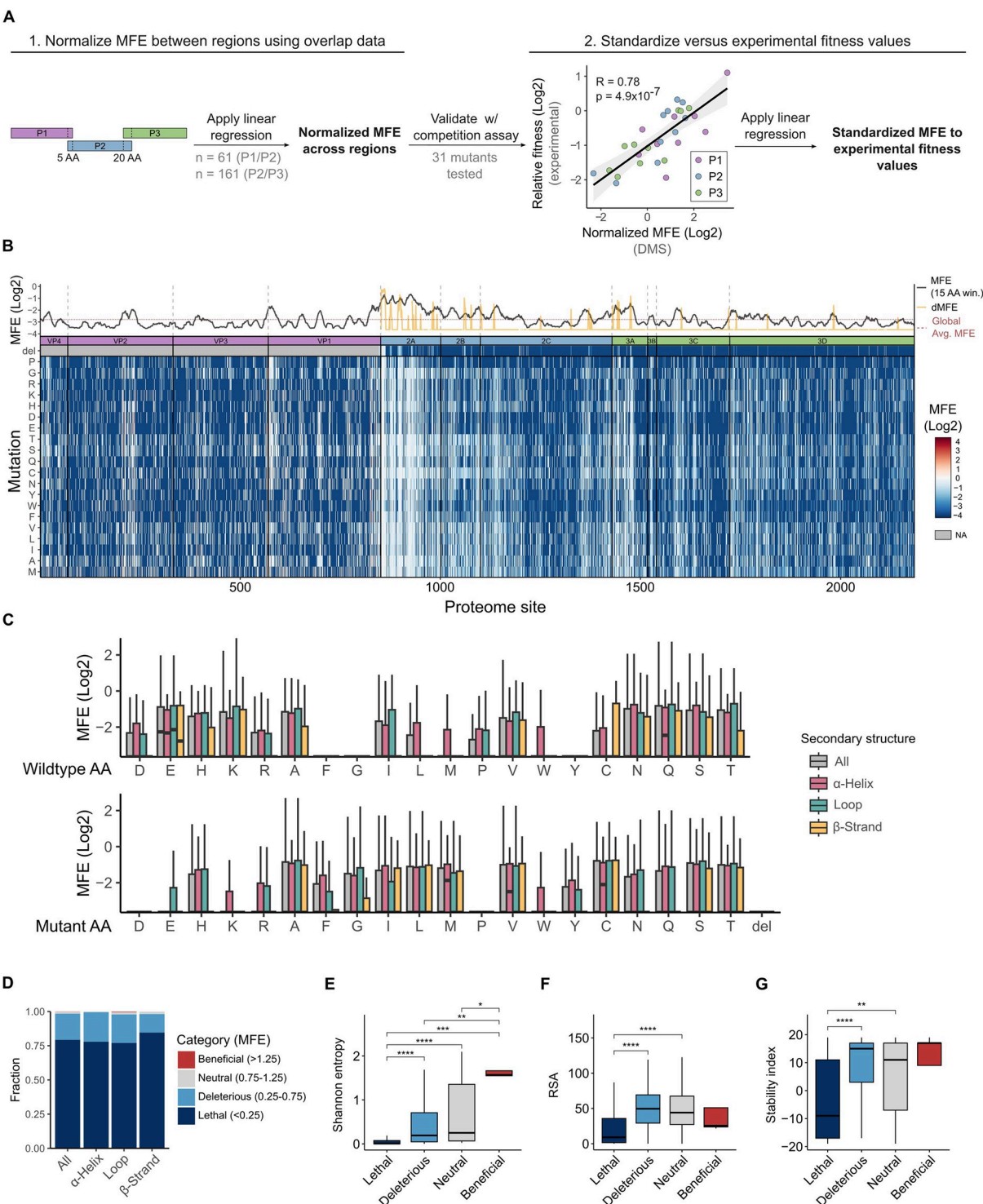

**Fig 2. Overall patterns of MFE across the CVB3 proteome.** (**A**) Overview of the MFE normalization protocol between the 3 mutagenized regions and their calibration to experimentally measured fitness values. A 5 and 20 AA overlap with the P1 and P3 region, respectively, was included in the mutagenized P2 region. Linear models based on the MFE of mutations in these overlapping regions were used to normalize the P1 and P3 MFE datasets relative to that of P2. These overlap-normalized MFE were then calibrated to experimentally measured fitness values of 31 individual mutants distributed across the proteome using a linear model, as these showed high correlation (Pearson correlation coefficient = 0.78, $p < 4.9 \times 10^{-7}$). (**B**) Proteome-wide MFE distribution. Top panel: line plot showing a 15 AA sliding window analysis of MFE (black line), individual deletion MFE (dMFE; yellow line), or the global average MFE across the full CVB3 proteome (dashed red line). Bottom: Heatmap of MFE and dMFE across the CVB3 proteome. (**C**) Boxplot representation of the distribution of MFE based on the nature of the WT

AA or the mutation relative to different secondary structure elements. Outliers (values outside 1.5× the interquartile range) are omitted for clarity. **(D)** The frequency of mutations belonging to lethal, deleterious, neutral, or beneficial fitness categories across the proteome (all) and according to the indicated secondary structure element. **(E–G)** The distribution of sequence variability in enterovirus B sequences (Shannon entropy) **(E)**, RSA **(F)**, or stability index **(G)** of each residue relative to different fitness categories. $p > 0.05$ not shown, $^*p < 0.05$, $^{**}p < 0.01$, $^{***}p < 0.001$, $^{****}p < 0.0001$ by Mann–Whitney test following multiple test correction. See S3 and S4 Tables for mutation and site data, respectively. See GitHub [19] section B4 for calculation and correction of MFE and section A4 for experimental data of the competition assay. The data underlying this figure can be found in S1 Data, page 1. AA, amino acid; CVB3, coxsackievirus B3; MFE, mutational fitness effect; RSA, relative surface area.

## Overall patterns of MFE across the CVB3 proteome

MFE varied across and between individual proteins, with several regions showing increased tolerance to mutations (e.g., N-terminal region of VP1, 2A, and 3A; Fig 2B and S3 Fig and S4 Table). Overall, the nature of the WT AA and the introduced mutation had a strong impact on MFE, which was further influenced by protein secondary structure for some AA. Specifically, mutations at 5 AA (M, F, G, W, and Y) had a strong fitness cost independent of the secondary structure, as did mutations from any AA to D, P, or a deletion (Fig 2C). The remaining AA showed differential MFE depending on secondary structure elements (e.g., mutation at L, M, and W in α-helices or mutation to E in loops; Fig 2C). Interestingly, the MFE of certain AAs varied depending on whether the residue was being substituted or if the mutation involved the AA itself. For example, F, G, and Y exhibited intolerance to mutations but were surprisingly tolerated as mutations themselves, likely reflecting their strict functional and structural roles as WT residues (Fig 2C). On the other hand, D and P displayed the opposite effect, showing a higher tolerance to accommodate substitutions than to be introduced as mutations (Fig 2C).

To facilitate the comparison of our results with evolutionary and structural parameters, MFE were grouped into fitness categories as follows: lethal (MFE of 0–0.25), deleterious (MFE of 0.25–0.75), neutral (MFE of 0.75–1.25), and beneficial (MFE >1.25; Fig 2D). Comparison of fitness categories with natural sequence variation (Shannon entropy) in human enterovirus B sequences ($n = 1,042$; collected from 1969–2023; see GitHub [19] section A6 for sequence information) revealed lethal mutations to concentrate in invariable residues while the opposite was observed for beneficial mutations, highlighting general agreement between lab-measured MFE and natural selection processes (Fig 2E). In line with this observation, the incorporation of MFE into phylogenetic models outperformed standard models (YNGKP/Goldman-Yang models) [20] for all 3 regions (S5 Table), revealing that DMS-derived MFE reflect selection processes in nature. As previously observed for the capsid region [18], multiple structural attributes were also correlated with MFE. Specifically, MFE differed by protein secondary structure, with loops having the largest fraction of neutral and beneficial mutations while β-strands had the largest fraction of lethal mutations ($p < 10^{-15}$ by Fisher's exact test for all; Fig 2D). In addition, lethal and deleterious mutations were enriched in buried residues (relative surface area, RSA, <25; Fig 2F) and had a larger fraction of destabilizing mutations (Fig 2G).

## MFE distribution within and between different protein classes

A general reduction in fitness was observed upon mutation across all CVB3 proteins (Fig 3A). However, clear differences in the ability to tolerate mutations were observed between individual proteins ($p < 10^{-16}$ by Kruskal–Wallis test; Fig 3A). The icosahedral CVB3 capsid is the most complex of all CVB3 protein structures. It is formed by the stepwise assembly of 60 copies of each of the 4 capsid subunits around the viral genome, with VP1-3 forming the external surface and VP4 lining the inner surface [21]. Of all CVB3 proteins, the structural proteins (VP1-VP4) had the largest fraction of lethal mutations but were the only ones harboring

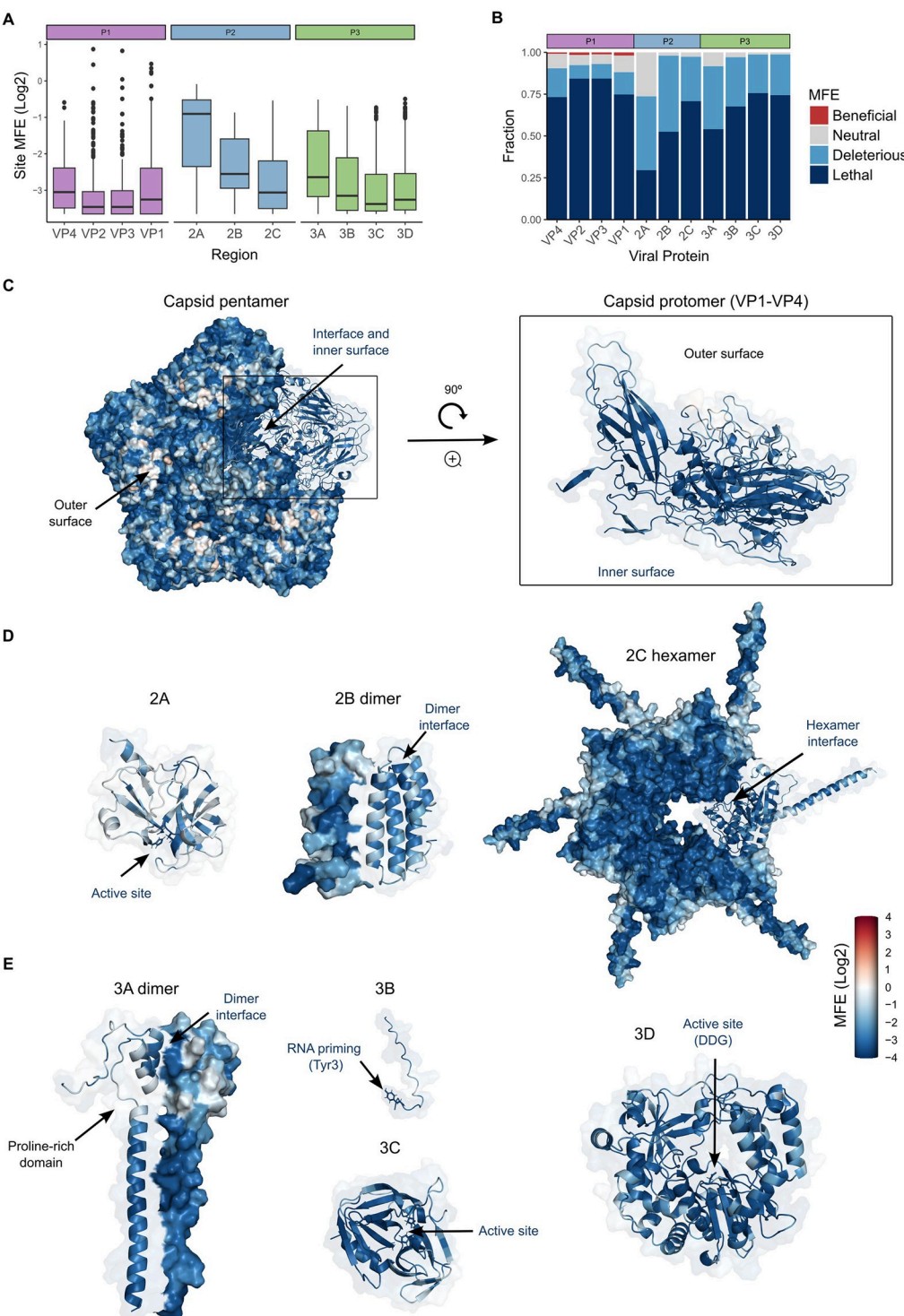

**Fig 3. MFE distribution across the CVB3 proteome. (A)** Boxplot of the average site MFE across all CVB3 proteins. **(B)** The distribution of fitness categories across the CVB3 proteins. **(C–E)** The CVB3 capsid pentamer **(C)**, P2-derived proteins **(D)**, or P3-derived proteins **(E)** colored according to the average MFE of each site. All P2 and P3 structures were predicted using Alphafold2 according to their known quaternary structure. For 2B, a predicted dimeric structure is shown as the tetramer prediction did not reflect the expected structure. See GitHub [19] section A7 for Pymol sessions of MFE mapped onto protein structures. The data underlying this figure can be found in S1 Data, pages 2 and 3. CVB3, coxsackievirus B3; MFE, mutational fitness effect.

beneficial mutations (Fig 3B and 3C). Overall, surface-exposed residues in the external capsid proteins (VP1-VP3) were more tolerant to mutations than buried residues (Fig 3C and S4A Fig), with beneficial mutations ($n$ = 230 mutations in 97 sites) enriched in surface-exposed loops (88.7%; $p < 10^{-9}$ by Fisher's test). The only sites in the proteome where mutations were beneficial on average (site MFE > 1.25; $n$ = 5) were found in the external capsid proteins. These sites belong to known antibody neutralization epitopes [22], likely reflecting the absence of this selective pressure in cell culture and highlighting its relevance to evolution in nature.

Viral proteases are responsible for the maturation of viral proteins from larger precursors and/or the cleavage of cellular factors to facilitate viral growth. CVB3 encodes both a chymotrypsin-like (2A) and a trypsin-like (3C) protease [23,24]. The 2A protease is only required for the cleavage at the P1/P2 junction to liberate the capsid precursor P1 [24,25], while 3C mediates the remaining 9 cleavage events at conserved Q-G/N residues [23]. Both proteases also cleave a non-overlapping array of cellular proteins to favor viral replication [23–26]. Comparison of MFE between these proteases revealed opposite mutation tolerance profiles, with 2A being the most, and 3C among the least, tolerant to mutations of all proteins (Fig 3A, 3B, 3D and 3E). In both proteases, mutations in the active site were highly deleterious (S4C Fig). However, a single mutation at each catalytic triad residue in 2A (H21Q, D39Q, C110W) or 3C (H40Q, E71D, C147W) was still observed in the viral populations at low frequency (maximal fitness of 0.19 and 0.32 for 2A and 3C, respectively; S4 Table), suggesting some minimal tolerance to mutations at these sites. As expected, mutations at the conserved Q residue of 3C cleavage sites were also highly intolerant to mutations (S4D Fig). However, Q to H substitution at these sites was also observed at low frequency in the viral populations (MFE < 0.25) in the nonstructural proteins but not in the capsid. This difference could be attributed to the nature of the protease mediating the cleavage, as the structural proteins are cleaved by the 3CD precursor rather than the mature 3C protease [27]. Alternatively, this could be related to the more complex maturation process of the capsid proteins, which requires the interaction with cellular protein folding factors that can dictate the fitness landscape of the capsid [28].

All positive-strand RNA viruses replicate on membranes and profound rearrangements of membranous compartments are frequently observed during infection [29]. In CVB3, the 2B, 2C, and 3A proteins have all been shown to alter cellular membranes by distinct mechanisms [30–33], although each protein also harbors additional activities. The CVB3 2B protein belongs to the viroporin family of oligomeric protein complexes [33]. Viroporins regulate ion permeability by inserting into membranes, where they form ion channels, and play multiple roles in viral replication and pathogenesis [33,34]. The 99 AA long CVB3 2B protein has an amphipathic α-helix connected to a second α-helix by a short loop [33] (see Fig 3D for a predicted dimeric structure). Overall, 2B showed a low tolerance to mutation, with few neutral mutations (2%) and a large fraction of deleterious (45%) and lethal (53%) mutations. The connecting loop was significantly more tolerant to mutation than both α-helices ($p < 10^{-6}$ by Mann–Whitney test; S4G Fig).

The viral protein 2C, or its precursor 2BC, can induce membrane rearrangements when expressed in cells [30]. However, 2C plays numerous additional functions in viral replication, including uncoating, RNA replication, genome encapsidation, and morphogenesis [32]. 2C is comprised of an N-terminal membrane-binding domain (MBD), a central ATPase domain of the Superfamily 3 AAA+ helicase family, a cysteine-rich domain (zinc finger domain), and a C-terminal helical domain. High-resolution structures have shown 2C to form a hexameric ring complex [35], which is essential for its ATPase activity. The different 2C domains showed strong variation in tolerance to mutations, with the MBD being the most accommodating of mutations and the ATPase domain the least (Fig 3D and S4I Fig). As expected for a multimeric protein, interface residues showed increased sensitivity to mutation ($p < 10^{-16}$ by Mann–

Whitney test; S4E Fig). The 2C region also encodes an RNA structure that is essential for genome replication, the *cis*-regulatory element (CRE). Mutations in this region incurred a higher cost to mutation than the remainder of the protein ($p < 10^{-15}$ by Mann–Whitney test versus the rest of 2C; S4F Fig), likely reflecting structural alterations to this key RNA structure.

The CVB3 3A protein disrupts cellular membranes by blocking anterograde transport between the endoplasmic reticulum (ER)-Golgi intermediate compartment (ERGIC) and the Golgi by preventing the formation of coatomer protein complex I (COPI)-coated transport vesicles [31]. It is a dimeric protein with an N-terminal domain (NTD) that is important for disrupting protein trafficking, an MBD, and a C-terminal domain (CTD). The 3A protein was the most tolerant to mutations following the 2A protease, with the NTD showing significantly higher tolerance to mutations relative to the MBD and CTD ($p < 10^{-16}$ by Mann–Whitney test; Fig 3E and S4H Fig). In agreement with this observation, mutations in the NTD that allow for viral replication without blocking membrane trafficking have been reported [32,36].

A defining feature of RNA viruses is the RNA-dependent-RNA polymerase (RdRP), which provides the essential function of replicating RNA genomes without utilizing a DNA intermediate. Overall, these proteins show a common fold that is conserved across both viral and nonviral RNA and DNA polymerases [37,38]. This includes a catalytic palm subdomain harboring the active site and catalytic GDD residues, finger domains that bind RNA, and the thumb domain that interacts with the dsRNA products. In the CVB3 RdRP (3D), the index finger showed the highest tolerance to mutation compared to all other polymerase domains (Fig 3D and S4J Fig). Interestingly, the index finger is also more variable in nature than the other domains, except for the thumb (S3 Table). As expected, all mutations in the catalytic GDD residues were invariably lethal, except for conservative substitutions of D329E and D330E that nevertheless resulted in a drastic reduction of fitness (>70% reduction versus WT). Finally, several positive-strand RNA viruses encode a small protein that is covalently attached to the viral genome (viral protein genome-linked; VPg) and is used to initiate replication by the RdRP, including picornaviruses, potyviruses, and caliciviruses. A tyrosine residue in these VPgs becomes uridylated, which provides a free hydroxyl that can be extended by the viral polymerase, enabling a primer-free replication mechanism [14]. As expected, all mutations at this critical residue in the CVB3 VPg (3B protein) were lethal (Fig 3E).

## Analysis of single codon deletions across nonstructural proteins

We next analyzed the effect of single codon deletions on viral fitness (deletion MFE; dMFE) across the nonstructural proteins. Overall, deletions were significantly more deleterious than mutations across both the P2 and P3 regions, with 96% of deletions reducing fitness below 25% of WT (Fig 4A and 4B). However, all nonstructural proteins harbored sites where deletions were not lethal (dMFE > 0.25; Fig 4B). As with MFE, the most tolerant region was found in 2A, with 27 sites maintaining viability upon deletion (S5A Fig). In addition, an 8 AA stretch at the N-terminus of 2A (residues 2–10) retained an average fitness of 82% (range, 68% to 100%; S5B Fig), likely reflecting flexibility in the 2A cleavage site sequence requirements. In agreement with this, 3 of the 5 residues in the C-terminus of VP1 that were included as part of the overlap with the P2 mutagenesis region also tolerated deletions (positions −5, −3, and −2 relative to AA 1 of P2; S4 Table). Additional residues with viable deletions were found in 2C and 3A ($n = 8$ each), 2B ($n = 4$), and 3C and 3D ($n = 2$ each; S5B and S5C Fig). Interestingly, nearly 4% of all deletions (51/1,305) were better tolerated than certain mutations at the same residue (dMFE > minimum MFE [minMFE]; Fig 4C) indicating that the structural and biochemical properties of some AA can be more deleterious than the absence of a residue, as previously observed [12]. Such sites were significantly more tolerant to mutations (Fig 4D) and

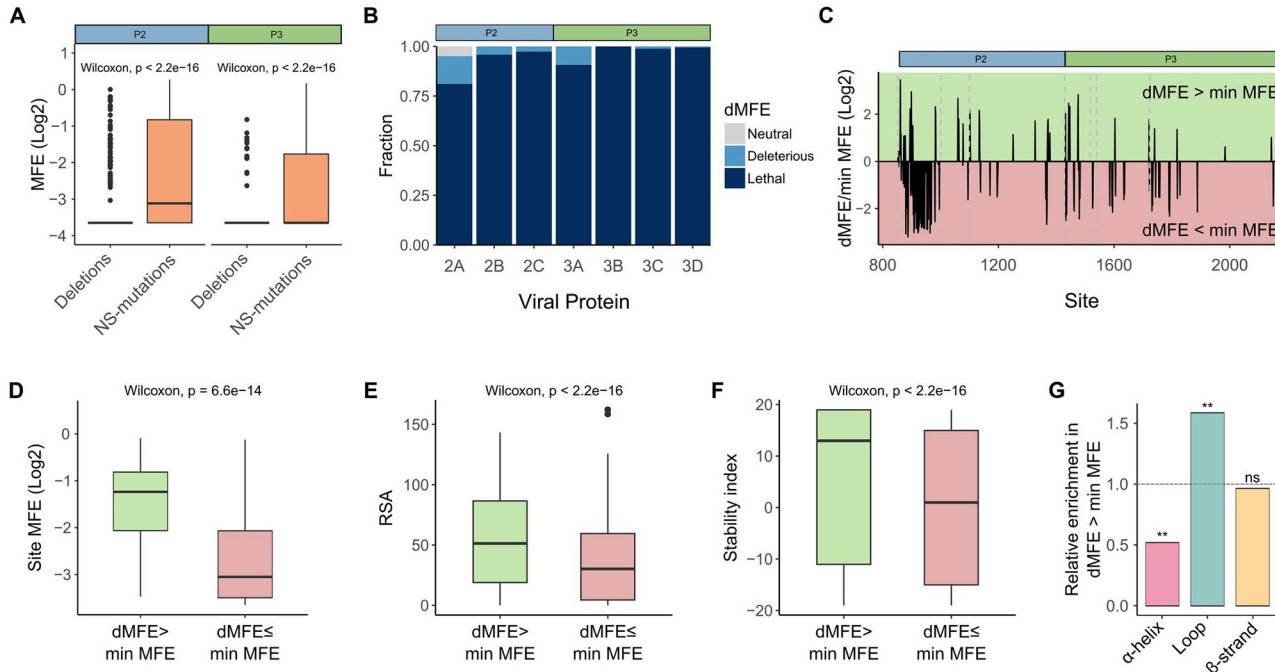

**Fig 4. Analysis of single codon deletions across the CVB3 nonstructural proteins. (A)** The effects of non-synonymous mutations (NS-mutations) versus deletions on viral fitness across the P2 and P3 regions. **(B)** The distribution of deletions according to their fitness categories in the nonstructural proteins. **(C)** Deletions can be better tolerated than mutations. The ratio of dMFE to the most detrimental MFE at each site (minMFE), with positive values indicating residues in which deletions were better tolerated than some mutations. **(D–G)** Deletions that are better tolerated than mutations (dMFE > minMFE) occur at sites that are more tolerant to mutations **(D)**, more surface exposed **(E)**, have a larger fraction of mutations that are stabilizing **(F)**, and show differential distribution between secondary structure elements **(G)**. ns $p > 0.05$, ** $p < 0.01$. The data underlying this figure can be found in S1 Data, pages 4–6. CVB3, coxsackievirus B3; dMFE, deletion mutational fitness effect; MFE, mutational fitness effect.

had specific structural attributes: high relative surface exposure (Fig 4E) and probability of stabilizing protein structure upon mutation (Fig 4F) as well as an enrichment in loops and depletion from α-helices (Fig 4G). The AA D and P were particularly enriched at such sites (>2.5-fold versus sites with dMFE < minMFE), where specific mutations were more deleterious than deletions (e.g., D to K, or P to F). Finally, these represented the majority of deletions having a nonlethal effect (approximately 73%; 37/51 of deletions with MFE >0.25).

## MFE remain constant across cellular environments

Viruses frequently infect different tissues during colonization of the host, which can present distinct environmental selection pressures [39]. To better understand whether MFE observed in HeLa-H1 cells are generalizable to different cellular environments, we obtained MFE across the full CVB3 proteome in an additional cell line, hTERT-immortalized retinal pigment epithelial (RPE) cells. These were chosen as they provide a significantly more restrictive environment for CVB3 than HeLa-H1 cells (116 ± 13-fold infectivity reduction) and mount a stronger type I interferon response (maximal inhibition of virus infection by 20 IU of interferon-β of 82 ± 7% and 22 ± 6% for RPE and HeLa-H1 cells, respectively).

To identify sites showing different fitness profiles between the 2 cell lines, RPE cells were infected with the mutagenized passage 1 virus populations from all 3 regions (P1, P2, and P3) produced in HeLa-H1 cells to obtain RPE-derived passage 2 virus populations (Fig 5A). Mutation frequencies in these populations were then derived using high-fidelity NGS as before and

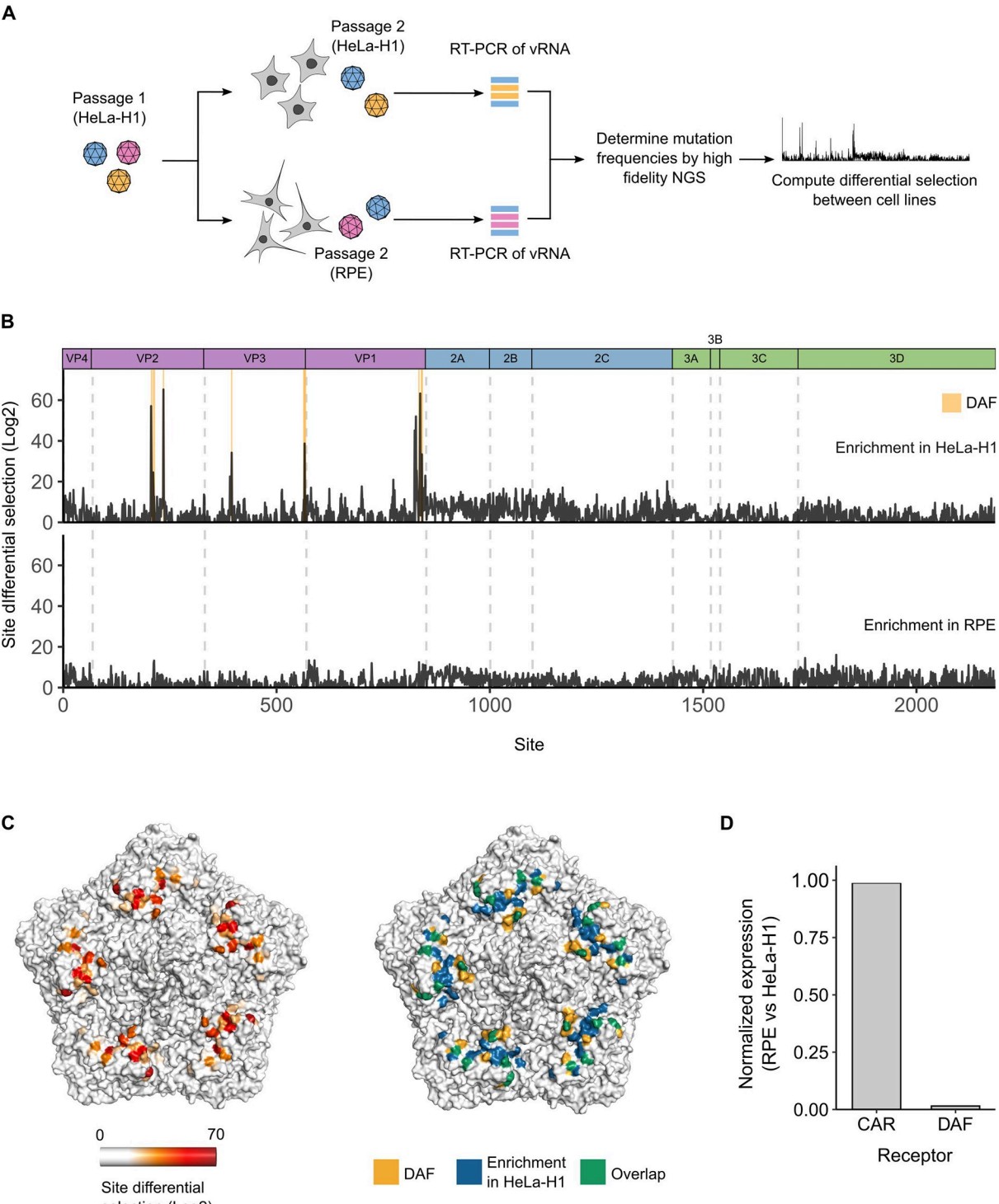

**Fig 5. MFE are conserved across cell lines.** (A) Overview of the experimental design for comparing MFE across HeLa-H1 and RPE cells. (B) Site differential selection for HeLa-H1 (top) and RPE cells (bottom). Residues showing large site differential selection scores that overlap with the DAF (CD55) footprint are highlighted in yellow. (C) The CVB3 capsid pentamer structure colored by the site differential selection score (left panel) or by sites showing large (>20) site differential selection scores in HeLa-H1, the DAF footprint, and their overlap (right panel). (D) Relative expression of DAF and CAR in RPE cells versus HeLa-H1 cells. Protein expression in each cell line was assessed by western blot and standardized to GAPDH expression. See S6 Table for NGS statistics and S7 for differential selection values between cell lines and S7 Fig for the western blots. The data underlying this figure can be found in S1 Data, pages 7 and 8. CAR, coxsackievirus-adenovirus receptor; CVB3, coxsackievirus B3; DAF, decay-accelerating factor; MFE, mutational fitness effect; NGS, next-generation sequencing; RPE, retinal pigment epithelial.

compared to those obtained in HeLa-H1 cells (see S6 Table for sequencing statistics and S6 Fig for correlations between replicates). For this, the relative frequency of each mutation compared to the WT AA at each site in viral populations derived from HeLa-H1 cells was divided by that observed in virus populations derived from RPE cells, yielding a differential selection score for each mutation (S7 Table). All mutations in each site showing enhanced fitness in HeLa-H1 cells (log2 differential selection score >1) were then summed to provide a site differential selection score that reflects the overall contribution of a site to improved fitness in this cell line (Fig 5B and S7 Table). Similarly, mutations with improved fitness in RPE cells (log2 differential selection score <1) were summed to reveal sites showing improved fitness in RPE cells (Fig 5B and S7 Table). Overall, sites across the nonstructural regions did not show strong differences in fitness between the 2 cell lines (Fig 5B), indicating similar selection pressures were present in both environments. In contrast, multiple residues across the capsid regions showed large fitness gains in HeLa-H1 cells (Fig 5B). Except for 1 mutation in the internal capsid protein, these sites localized to surface-exposed loops, suggesting a role in receptor binding. As CVB3 can utilize 2 receptors for cell entry, the coxsackievirus-adenovirus receptor (CAR) and decay-accelerating factor (DAF; CD55) [40,41], we examined whether these sites colocalized with the binding footprint of either receptor. Indeed, most sites overlapped with DAF-binding residues (Fig 5B and 5C). In support of this, while CAR expression was similar between both cell lines, DAF was only detected by western blot in HeLa-H1 cells (Fig 5D and S7 Fig). Hence, cell-specific entry factors impose strong selection pressures between different cell lines.

## Identification of mutation-intolerant druggable pockets

The extreme mutation rates and large population sizes of RNA viruses generate tremendous diversity during infection. As a result, antiviral therapies can be rendered inefficacious by the rapid selection of variants encoding drug-resistant mutations that maintain sufficient replicative fitness to spread in the host [42]. A potential means of preventing the emergence of drug resistance is the targeting of druggable pockets that accommodate few viable mutations. To evaluate if different druggable pockets show distinct mutation tolerance profiles, we combined computational druggable pocket predictions with our MFE information across the CVB3 proteome (Fig 6A). The SiteMap software [43], which identifies druggable pockets based on different structural and physicochemical properties (e.g., exposure and enclosure, contact properties, hydrophobic and hydrophilic character), was used to predict druggable pockets in the CVB3 proteome. In total, SiteMap identified 12 pockets with a high probability of being druggable (SiteScore>0.8 [43], see Methods). These were distributed across the capsid (*n* = 5), 2A (*n* = 1), 2B (*n* = 3), 2C (*n* = 1), and 3D (*n* = 2; S8 Table) proteins. Mapping of MFE to these druggable pockets revealed strong variation in mutational tolerance (Fig 6B and S8 Table). Specifically, drug pocket 12 in the 3D polymerase was highly intolerant to mutation, having only a single neutral mutation out of 679 possible mutations in its 34 residues, with the remaining mutations being either deleterious (14%) or lethal (86%; Fig 6B and 6C). In contrast, druggable pocket 6 in 2A had 54 neutral or beneficial mutations across 18 of its 24 residues (75%), suggesting a lower barrier to the development of drug resistance (Fig 6B and 6C). Similarly, all druggable pockets in the capsid had a significant fraction of neutral mutation (3.1% to 7.9%), with 80% of them even having beneficial mutation (0.4% to 1.2% of all mutations; Fig 6B). Hence, the incorporation of MFE data into antiviral target selection could help identify drug targets with higher barriers to drug resistance.

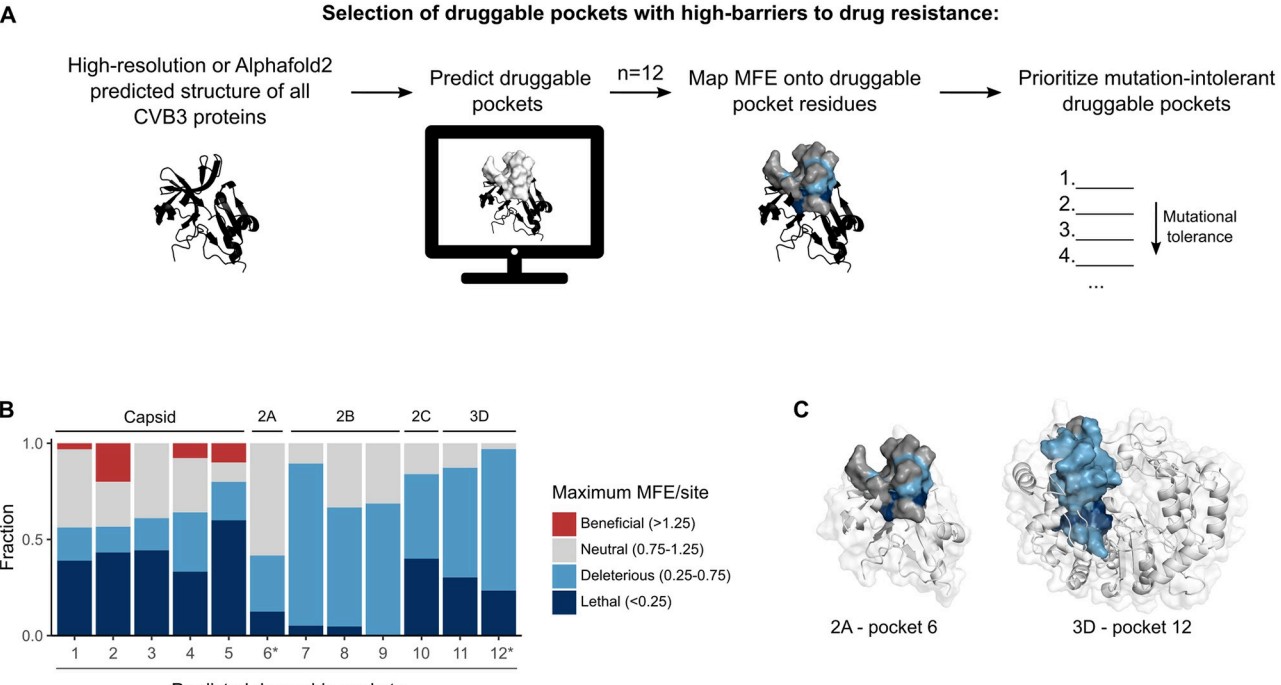

**Fig 6. MFE can inform druggable pocket selection. (A)** Overview of the pipeline for prioritization of druggable pockets based on MFE data. **(B)** The fraction of residues in each drug pocket belonging to a given fitness category based on the most permissive mutation at each residue (i.e., maximum MFE per site). Asterisks indicate drug pockets mapped onto the protein structure in (C). **(C)** Mapping of pocket 6 and 12 residues onto the 2A and 3D protein structure, respectively, and colored according to the fitness category of the most permissive mutation at each residue. See S8 Table for data on druggable pockets and MFE. MFE, mutational fitness effect.

## Conclusions

The average rates of nucleotide misincorporation by the polymerases of RNA viruses are generally orders of magnitude higher than those observed for other organisms due to the absence of proofreading mechanisms [2–4]. These extreme mutation rates are believed to in part be the result of a tradeoff between replication kinetics and fidelity [44], which RNA viruses have optimized to increase replication speed while avoiding population extinction due to mutational load. In addition, high mutation rates confer RNA viruses the mutational diversity required to rapidly evade immunity, develop drug resistance, and adapt to new hosts. As most mutations are deleterious, it has been suggested that viral proteins harbor unique structural properties to mitigate mutational fitness costs [45]. However, mutational diversity in nature is unevenly distributed across and within different viral proteins [7–9], suggesting distinct mutational tolerance for different viral protein classes. Indeed, RNA viruses encode a diverse set of protein classes, which vary in size (22–462 AA in CVB3) and complexity (1–240 subunits in CVB3), which could influence mutational tolerance.

A comprehensive analysis of how mutations affect fitness across different protein classes has not been reported. An analysis of how approximately 20% of all possible single AA mutations affected poliovirus fitness revealed differences in the overall distribution of MFE between structural and nonstructural proteins, with the latter having a higher fraction of neutral and positive mutations [10]. However, this work largely focused on the effects of single mutations per codon, which limits the mutational space sampled, and did not analyze different proteins in detail. Rather than relying on the viral polymerase to generate mutations, DMS

experimentally introduces a large fraction of all possible single AA mutations across proteins of interest. As such, it is ideal for obtaining a comprehensive understanding of MFE. Thus far, this technique has been used to define MFE in individual proteins, including structural proteins from enveloped and non-enveloped viruses or nonstructural proteins such as polymerase or matrix proteins [13,17]. However, a full proteome DMS analysis that compares mutation tolerance between different viral proteins has yet to be reported.

In this work, we utilized mutagenized viral populations from a previous study of the CVB3 capsid region [18] with a new mutagenesis protocol of the nonstructural proteins to provide a comprehensive analysis of MFE across a full proteome. Importantly, we calibrated the DMS-derived MFE relative to experimental measurements of viral fitness to provide a more realistic measure of fitness that is lacking in the majority of published DMS studies, including our previous analysis of the CVB3 capsid. Overall, we define the effect of >97% of all possible single AA mutations across the proteome of CVB3. Aside from providing a wealth of information with which to interpret the biology and evolution of CVB3 and related viruses, our results reveal both general and specific information on how different protein classes accommodate mutations. Firstly, a strong bias in MFE is observed depending on both the original and mutant AA, which was further influenced by their presence in different secondary structure elements for certain AA (Fig 2C). Moreover, even for the same AA in the same secondary structure element, MFE could vary strongly depending on whether a given AA was being mutated from (i.e., the WT residue) or to (i.e., the mutation), highlighting strong structural and functional constraints (Fig 2C). As expected, external residues in the capsid and surface-exposed residues in the nonstructural proteins were more tolerant of mutations, highlighting a general sensitivity of buried residues to mutation (S4B Fig). In terms of protein classes, the structural proteins showed a higher fraction of lethal mutants than the nonstructural proteins but were also the only ones to have beneficial mutations (Fig 3A and 3B). Interestingly, the only sites showing an average positive effect of mutation on fitness were found at antibody neutralization sites. This suggests that immune evasion constitutes an important selection pressure in nature, one which comes at a cost to viral fitness. Of the nonstructural proteins, the 2 proteases showed different MFE profiles, with 2A being the most tolerant to mutations of all proteins and 3C among the least (Fig 3A and 3B). While this difference could be due to relaxed selection pressures in cell culture, 2A is also more variable than 3C across enterovirus B sequences ($p < 10^{-16}$ by Mann–Whitney test; S3 Table), supporting inherent structural and functional differences. As 2A only performs a single cut in the CVB3 proteome, versus nine performed by 3C, this protease could be less constrained. Indeed, the protease function of 2A has been replaced by a short ribosomal skip-sequence in some picornavirus genomes, while in others it has been duplicated [24]. The remaining nonstructural proteins showed varying sensitivity to mutation, with particular regions and/or domains showing differential ability to accommodate mutations that were associated with protein-specific functions.

Unlike mutations, experimental evaluation of the effects of deletions on protein function has only recently begun to be studied [5,11–13]. Taking advantage of the new synthetic biology mutagenesis approach utilized in this work, we included an analysis of all single AA deletions across the nonstructural proteins together with all single AA mutations. In agreement with prior work [5], we observed deletions to be more deleterious than mutations, with the vast majority being lethal (Fig 4A and 4B). However, some deletions were neutral in the 2A protein, the most mutation-tolerant CVB3 protein (Fig 4C). These were concentrated at the N-terminus of 2A, where this protease cleaves the upstream capsid region, highlighting flexibility in the sequence requirements of this protease. Interestingly, nearly 4% of deletions had higher fitness than the most deleterious mutation at the same site, as previously reported for

the receptor-binding domain of the SARS-CoV-2 Spike protein [12]. The larger number of mutations included in our study allowed us to identify specific characteristics of these sites, which were generally more tolerant to mutations, harbored a larger fraction of stabilizing mutations, and were enriched in surface-exposed loops. However, whether they play particular structural or functional roles, or impact viral evolution, remains to be defined. In support of the latter, a recent comprehensive analysis of codon deletions and insertions (indels) in the related Enterovirus A71 showed these to vary across the Enterovirus A species, suggesting a role for indels in shaping the evolutionary trajectories of these viruses [11]. Additionally, in agreement with our results, this study found the N-terminal regions of 2A and 3A to be tolerant to deletions, suggesting tolerance to deletions to be generalizable across different enterovirus species.

The majority of DMS studies analyze MFE in a single, highly susceptible cell line. Whether the derived MFE are generalizable to additional cellular environments, and in particular to more restrictive environments, is unclear. To address this, we compared MFE between HeLa-H1 cells and a less-permissive, immune-competent cell line, RPE cells. Interestingly, MFE remained relatively constant across most sites in the proteome, indicating similar selection pressures across the different cellular environments (Fig 5B) [21,43]. However, several capsid sites showed improved fitness in HeLa-H1 cells compared to RPE cells, which correlated with the expression of the DAF coreceptor in HeLa-H1 cells (Fig 5C and 5D, and S7 Fig). Hence, entry factors, which can show differential expression across different tissues, can impart strong selection pressures on viruses. It is important to note that any mutations with low fitness in HeLa-H1 but high fitness in RPE cells may have been selected against during the first passage in HeLa-H1 cells and may therefore be missing in our experiments.

Antiviral drug development is a long, complex, and costly process. Unfortunately, the extreme evolutionary capacity of RNA viruses can rapidly render antiviral therapies ineffective [42]. A potential application of DMS can be the identification of protein regions that are intolerant to mutations and can therefore present a higher barrier to the emergence of drug resistance. As a first estimation of whether MFE can be used to inform drug target selection, we incorporated our MFE data into predicted drug pockets across the CVB3 proteome. This revealed a highly intolerant druggable pocket in 3D, where only a single mutation out of all possible mutations across the 34 pocket residues had a neutral MFE, with the remaining mutations being either deleterious or lethal. Such a drug pocket is likely to present a strong barrier to the development of resistance due to the high fitness costs incurred by drug-resistant mutations. On the other hand, druggable pockets in the capsid and the mutation-tolerant 2A protein had a notable fraction of neutral mutations suggesting a low barrier to the development of drug resistance. Beyond prioritizing drug targets, MFE data can be used to identify mutations that are likely to arise in a given druggable pocket due to low fitness costs. These can then be incorporated into in silico drug screening to a priori select compounds that will be resistant to such mutations and/or to inform medicinal chemistry. Of note, the DMS approach used in our work is limited by its use of single AA mutations, which precludes the analysis of compensatory mutations that can restore the fitness of deleterious mutations. However, profiling drug resistance using DMS methodologies that evaluate multiple mutations is feasible [46], in particular when focusing on a limited protein region such as a druggable pocket. In addition, escape from antivirals can also be conferred by residues that are external to the drug-binding pocket (e.g., [35]), requiring more complex structural bioinformatic approaches that assess the contribution of nearby residues on the drug pocket.

## Materials and methods

### Cells, viruses, and reagents

HeLa-H1 (CRL-1958; RRID: CVCL_3334) and RPE (CRL-4000) cells were obtained from ATCC. Both cell lines were periodically validated to be free of mycoplasma. Cells were cultured in culture media (Dulbecco's modified Eagle's medium, Pen-Strep, and L-glutamine) supplemented with 10% or 2% heat-inactivated fetal bovine serum for culturing or infection, respectively. The human codon-optimized T7 polymerase plasmid was obtained from Addgene (#65974) and the CVB3 infectious clones encoding mCherry (CVB3-mCherry) or eGFP (CVB3-eGFP) were previously described [47]. The titer of these reporter viruses was obtained by infecting HeLa-H1 cells with serial dilutions of each virus in 96-well plates and counting the number of fluorescent cells at 8 h postinfection using an Incucyte SX5 Live-Cell Analysis System (Sartorius). All virus experiments were carried out under BSL2 conditions after obtaining approval from the biosafety committees of both I2SysBio and the University of Valencia. All work with genetically modified organisms was approved by the relevant national committees. Antibodies targeting CAR (E-1; sc-373791), CD55/DAF (NaM16-4D3; sc-51733), GAPDH (0411; sc-47724), and goat anti-mouse IgG-HRP (sc-2005) were purchased from Santa Cruz and interferon-β was purchased from Abcam (ab71475).

### Mutant library construction

P1 DMS libraries were previously described [18]. Briefly, the full capsid region of CVB3 was subjected to a PCR-based mutagenesis protocol using degenerate primers, with random bases at the codon-matching positions (NNN). The mutagenized region was cloned into the CVB3 infectious clone and used to generate viral populations, which were previously characterized in detail [22]. To generate the P2 and P3 mutagenized regions, a synthetic biology approach was utilized. Briefly, each region was divided into approximately 250-bp adjacent regions for the mutagenesis and was flanked by complementary sequences to the upstream and downstream regions in the infectious clone. In total, 8 and 10 such tiles were designed for P2 and P3, respectively. Within each tile, 20 oligonucleotides were ordered for each residue, 19 encoding each of the possible AA mutations, and 1 with the WT codon deleted. Mutated codons were designed to incorporate the maximum number of nucleotide changes versus the wild-type codon and the more frequently utilized codon in the CVB3 ORF was chosen when multiple possibilities were available. In addition, stop codons were included at GTG, GAA, AAA, TAC, CAA, and GAG codons, and synonymous codons were included at R, L, and S AA as controls. Oligonucleotides were ordered from Twist Biosciences as a single pool for the P2 region and as 2 pools for the P3 region (see GitHub [19] section A2 for codon choices and full tile sequences). Each tile region was then amplified by PCR using primers matching the overlap region with KAPA HiFi DNA polymerase for 14 to 16 cycles. PCR products were purified and size-selected using CleanNGS (CleanNA) beads (ratio of 0.75×) and concentrated using DNA clean and concentrator columns (ZYMO research). Finally, the purified PCR was joined to a complementary PCR of the infectious clone using the NEBuilder HiFi DNA assembly kit (New England Biolabs; see S1 Table for primer sequences). In total, 3 libraries were generated for each tile. Due to the limited quantity of synthetic oligonucleotides, the third replicate of the P2 libraries was amplified by PCR as described above and cloned into the pJET plasmid using the CloneJET PCR cloning kit (Thermo Fisher Scientific) for downstream amplification. Plasmid-cloned tiles were amplified for 14 cycles by KAPA HiFi, size-selected using E-Gel SizeSelect II Gels (Thermo Fisher Scientific), and purified using CleanNGS (CleanNA) beads (1×). Tiles were cloned into the CVB3 infectious clone using NEBuilder HiFi DNA assembly as previously described [18].

The assembled plasmid reactions were purified using a Zymo DNA Clean and Concentrator-5 kit (Zymo Research) and electroporated into bacteria as previously described [18]. Cells were then grown overnight in a 10-ml liquid culture at 37°C and DNA purified using the NZY-Miniprep kit (NZYtech). Transformation efficiency was estimated by plating serial dilutions of the transformation on agar-amp plates. In total, $6–18*10^5$ transformants were obtained for each line. Viral genomic RNA was then in vitro transcribed and electroporated into HeLa-H1 cells as previously described [18]. Electroporated cells were then pooled and cultured for 9 h to produce the passage 0 virus (P0). Following 3 freeze–thaw cycles, cell debris were removed by centrifugation and viral titer was obtained by plaque assay. Two additional rounds of passaging were performed in HeLa-H1 cells using $10^6$ plaque-forming units (PFUs; multiplicity of infection, MOI, of 0.1) and allowing infection to continue for a single cycle (8 h). All infections in HeLa-H1 cells produced $>1.30 \times 10^7$ PFU in P0 and $>2.45 \times 10^7$ PFU in P1 as judged by plaque assay. For RPE cell infection, the passage 1 virus populations produced above were used to infect $5 \times 10^6$ cells for 8 h at an MOI of 0.1 by using $1 \times 10^7$ PFU as determined using a plaque assay in HeLa-H1 cells to compensate for the lower susceptibility of RPE cells.

## Duplex sequencing

Viral RNA extracted as indicated above was reverse transcribed with the high-fidelity One-Script Plus Reverse Transcriptase (Applied Biological Materials) using 8 μl of RNA and the primers P2_RT_5195 (AATGAAAGCCCGACTGACATGTTT) or CV_7414_R (TTTTTTTTTTTTTTTTCCGCAC) for P2 or P3 regions, respectively. The number of copies was quantified via qPCR using PowerUp SYBR Green Master Mix (Thermo Fisher Scientific) in a 10 μl total reaction, using 1 μl of the template and the primers CVB3P1_qPCR_Fwd (GGAAG-CACGGGTCCAATAAA) and CVB3P1_qPCR_Rev (CAGAGTCTAGGTGGTCTAGG-TATC) for the P2 region or CVB3P2_qPCR_F (ATGGCTGCCCTAGAAGAGA) and CVB3P2_qPCR_R (CTGACACGGTTGGAGCATTA) for the P3 region. A standard curve was generated using the CVB3 infectious clone plasmid. The full region of interest was then amplified from $>10^6$ copies of cDNA using Phusion polymerase (Thermo Fisher Scientific) and the primers CVB3_P2_seq_F (AACGTGAACTTCCAACCCAGC) and CVB3_P2_seq_R (CGATTTGAGCAGGTCCGCAAT) to amplify the P2 region or the primers CVB3_P3_seq_F (TCTTGTGTGTGGGAAGGCTATACAAT) and CVB3_P3_seq_R (ACCCCTACTG-TACCGTTATCTGGTT) to amplify the P3 region. Duplex sequencing libraries were prepared as previously described [18] with some modifications. New adaptors were developed that reduce random tag length to 8 and enable dual indices to eliminate index hopping and a synthetic fragment was used to calibrate the amount of DNA used for generating the libraries (see supplementary protocol in GitHub [19], section A1). Samples were sequenced on an Illumina Novaseq 6000 sequencer. The resulting files were analyzed as previously described [22] using a modified version of the duplex sequencing pipeline [48], except that a modification was introduced into the VirVarSeq script [49] to enable the identification of codon deletions (full analysis scripts including the modified VirVarSeq script are available on GitHub [19], section B2) and the 2021 version of the duplex UnifiedConsensusMaker.py script was used (https://github.com/Kennedy-Lab-UW/Duplex-Seq-Pipeline; minimum family size of 2 and 70% agreement within family required to call a mutation). With this, the counts of each codon at each position were obtained (codon tables, available on GitHub [19], section A3).

## Calculation and visualization of MFE and differential selection

As before [18], all single mutations in codons were omitted from the analysis to increase the signal-to-noise ratio in the case of P1. For P2 and P3, codon tables were filtered to keep only

those codons where mutations were introduced. MFE were calculated following the general procedure described in dms_tools2 for the ratio method [50] (https://jbloomlab.github.io/dms_tools2/index.html) using custom scripts (available on GitHub [19], section B3). Briefly, for each site, the sum of all codons giving rise to a particular AA mutation was divided by that of the WT AA to obtain the relative enrichment of each mutation at each site. To obtain MFE, the relative enrichment for a given mutation in the viral populations was divided by its relative enrichment in the mutagenized libraries. To avoid zeros in the numerator, a coverage-scaled pseudocount of 1 was added to the count of each mutation. Additionally, all mutations which were not observed at least 5 times in the mutagenized libraries were omitted from analysis and mutations observed at least 5 times in the mutagenized libraries and 0 times in the viral populations were defined as lethal. Finally, the log2 MFE from replicate lines were then averaged to obtain the average MFE of each mutation, as long as at least 2 MFE values were observed among the independent replicates. To normalize MFE between regions, MFE belonging to the overlap regions were used to fit a linear model with the R lm() function. These were then applied to all MFE from the corresponding region (P1 or P3; see GitHub [19], section B4 for data and scripts). Overlap-normalized MFE were used to obtain a linear model for DMS-derived MFE relative to experimental fitness measurements following outlier removal with the Cook's Distance method (threshold for outliers: Cook's distance > 4 / the number of observations). This transformation was applied to the full dataset (see GitHub [19], section B4 for data and scripts). For downstream analyses, lethal mutations were assigned the minimum MFE value observed in the full proteome. All MFE plots in the main figures represent the mean MFE for each mutation at each site across replicates. Line plots represent the mean MFE per site.

To calculate differential selection between HeLa-H1 and RPE cells, the relative enrichment of a given mutation in the HeLa-H1 viral populations was divided by its relative enrichment in the RPE viral populations. As indicated above, a coverage-scaled pseudocount of 1 was added to the count of each mutation and, in this case, mutations that were not observed at least 5 times in one of the 2 viral populations were omitted from the analysis. The log2 differential selection scores from replicate lines were then averaged to obtain the average differential selection of each mutation, as long as at least 2 differential selection values were observed among the independent replicates. All mutations having a mutation differential selection score >1 or <1 were summed to define the site differential selection score for HeLa-H1 or RPE cells, respectively. Scripts to perform the differential selection analysis are available on GitHub [19] (section B5).

## Experimental assessment of viral fitness

CVB3 mutants were generated by site-directed mutagenesis of the mCherry-CVB3 fluorescent infectious clone as previously described [22] and utilized in competition assays as previously described [22]. Briefly, all mutants were recovered in HEK293T cells by transfection of linearized DNA together with a T7 polymerase expression plasmid. These mutants or the WT mCherry reporter virus were mixed at a 1:1 ratio with a GFP-expressing CVB3 reference virus and used to infect HeLa-H1 cells at an MOI of 0.001 in a 24-well plate. The number of cells infected with each virus (GFP+ or mCherry+) was determined at both 8 hpi (initial infection ratio) and 20 hpi (following approximately 2 rounds of replication) using a live cell microscope (Incucyte SX5; Sartorius). The relative fitness of each mutant or the WT mCherry virus was calculated relative to the GFP-expressing reference virus using the formula ($\text{mCherry}^+_{20h}$/$\text{GFP}^+_{20h}$)/($\text{mCherry}^+_{8h}$/$\text{GFP}^+_{8h}$). Fitness values were then standardized relative to that of the WT virus. The results of the competition assay are available on GitHub [19] (section A4).

## Cell-susceptibility, interferon sensitivity, and receptor expression analyses

For calculation of cell susceptibility, HeLa-H1 and RPE cells were infected with identical quantities of an mCherry-expressing CVB3, and virus titer was determined by examination of fluorescence at 8 h postinfection on a live-cell microscope (Incucyte SX5; Sartorius). For evaluation of type I interferon sensitivity, cells were mock treated or treated with 20 IU/ml of interferon-β (ab71475; Abcam) for 24 h. Subsequently, the cells were infected with CVB3-mCherry at a low MOI in triplicate. The number of infected cells was quantified by examining mCherry+ cells at 14 hpi using a live cell microscope (Incucyte SX5; Sartorius). For analysis of receptor expression, $5 \times 10^6$ cells were lysed using 0.5 ml of lysis buffer (50 mM HEPES (pH 7.2), 1% NP-40) containing protease inhibitor cocktail (78430; Thermo Fisher Scientific) on ice for 5 min. Nuclei were then separated by centrifugation (10,000 × g for 5 min) at 4˚C. The lysates were mixed with Laemmli buffer containing β-mercaptoethanol (1610747; Bio-Rad), denatured at 95˚C for 5 min, separated on a 4% to 20% gradient precast gel (4561095; Bio-Rad), and transferred to a PVDF membrane. Subsequently, the membrane was blocked with TBS-T containing 3% of BSA, incubated with the relevant primary antibodies for 1 h (anti-DAF: SC-51733, anti-CAR: SC-373791, anti-GAPDH: SC-47724; Santa Cruz Biotechnology), washed 3 times with TBS-T and incubated with the secondary antibody for 1 h (anti-mouse-HRP: SC-2005; Santa Cruz Biotechnology). Finally, the membrane was washed 3 times, and the chemiluminescent substrate was added (34577; Thermo Fisher Scientific). Quantification of protein bands was done with ImageJ.

## Bioinformatic analyses

For examination of sequence variability in enterovirus B alignments, all available sequences matching the criteria "taxon id 138949, minimum size 2180, 0 ambiguous characters, host homo sapiens, and exclude lab host," were downloaded from NCBI virus on February 8, 2024. Sequences were then filtered in R to remove those with non-ATGC characters ($n$ = 1,042 remaining), translated to AA sequences using the translate function from the Biostrings package (Version 2.70.1), and aligned to the CVB3 Nancy ORF using the AlignSeqs function of the DECIPHER package (version 2.30.0). All positions with gaps in the lab Nancy CVB3 strain were removed from the alignment (see GitHub [19], section A6 for alignment), and AA counts at each position were summed using the ConsensusMatrix function of Biostrings. Finally, Shannon's entropy of AA was calculated using the formula -sum(log(p[p>0]/sum(p)) * (p[p>0]/sum(p))), where p is the frequency of a given AA. Phydms [20] was used to evaluate if the incorporation of MFE into phylogenetic models could improve model fit compared to standard models. For this, all available CVB3 sequences were downloaded on February 8, 2024 using the abovementioned criteria but with taxon id 12072. Sequences were aligned and processed as indicated above for entropy measurements and split into the P1, P2, or P3 regions based on the location of the lab reference strain. MFE values were transformed into site preferences by first setting all missing values to the average MFE at a particular site and normalized by dividing by the sum of all MFE values at each site. Phydms was then run using the default setting following preparation of the sequence alignment with the phydms_prepalignment function (see GitHub [19], section A6 for subalignments and AA site preferences).

The CVB3 capsid structure was based on the PDB:4GB3, but modified to match the Nancy sequence as previously reported [18]. Structure prediction for all remaining CVB3 proteins was performed using Alphafold [51] (version 2.0.1), with parameters max_template_-date = 2020-05-14 and uniclust30_2018_08. The multimer version of Alphafold [52] was similarly run for all multimeric proteins using the relevant number of subunits. For 2B, the dimeric prediction yielded more similar results to the expected structure [33,53,54] than the

tetrameric prediction and was used. Protein stability calculations were obtained from a prior analysis for the capsid proteins [18] or performed on the Alphafold2 predicted structures using MutateX [55]. To calculate the Stability index, the number of mutations predicted to be neutral or stabilizing (predicted ΔΔG<1) at each residue was subtracted from the number of mutations predicted to be destabilizing (predicted ΔΔG>1). RSA and secondary structure were obtained from PSAIA [56] and STRIDE [57], respectively, using the Alphafold2 predicted protein structures. Residues with RSA>0.25 were classified as surface exposed. Interface residues were obtained using the interfaceResidues script in PyMOL (cutoff of interface accessible surface area = 0.75 Å$^2$). Identification of druggable pockets was performed using the Schrödinger software (Version 13.8). Proteins were first subjected to the *protein preparation workflow* in Maestro using the default settings and then SiteMap was used to define druggable pockets using the default settings, with sites having a SiteMap score >0.8 considered as druggable according to previously suggested criteria [43].

### Statistical analyses

All statistical tests were performed in R (version 4.3.2) and were two-tailed. Multiple testing corrections were performed using the FDR method.

### Supporting information

**S1 Fig. Characterization of the DMS libraries by Sanger sequencing.** A total of 59, 148, and 92 clones were sequenced for the P1, P2, and P3 region, respectively. The fraction of each number of codons mutated per clone (**A**) and each type of mutation (**B**) are graphed. Of note, the full capsid region was sequenced for P1, while for P2 and P3, only the corresponding mutagenized tile region was sequenced. The data underlying this figure can be found in S1 Data, page 9.
(TIFF)

**S2 Fig. Correlation of mutation, site, and deletion MFEs in HeLa-H1 cells between replicates.** (**A**) Correlations matrices for MFE of mutations (mut MFE), their average per site (site MFE), and deletions (dMFE) for independent replicate lines (L1-L3) for the P1, P2, and P3 regions. Of note, for P1, only 2 replicates were used and deletions were not included in the mutagenesis protocol, precluding their analysis.
(TIFF)

**S3 Fig. Linear models used for the normalization of MFE between the different mutagenized regions.** Graphical representation of the linear model (black line) and 95% CI (gray shade) obtained for the MFE of mutations present in the overlap region between P1 and P2 (R$^2$ = 0.54, *p* = 7.338 × 10$^{-12}$) (**A**) or P2 and P3 (R$^2$ = 0.69, *p* < 2.2 × 10$^{-16}$) (**B**) used for normalization between regions. The Pearson correlation coefficient and associated *p*-value are shown. The data underlying this figure can be found in S1 Data, page 10.
(TIFF)

**S4 Fig. Protein-specific features of MFE. (A, B)** Distribution of site MFE in outer surface residues versus other residues in the capsid proteins VP1-VP3 (**A**) and in surface-exposed residues versus internal residues for the nonstructural proteins (**B**). (**C, D**) Distribution of MFE in the active sites of the 2A and 3C proteases, and the 3D polymerase versus all other residues in each protein (**C**), and in the 3C protease cleavage site Q residues of each protein versus all other Q residues in that same protein (**D**). (**E**) Distribution of MFE in interface residues between monomers versus other residues in 2C. (**F**) Distribution of MFE in the CRE element versus other residues in 2C. (**G–J**) Distribution of MFE in the different structural and

functional domains of the 2B **(G)**, 3A **(H)**, 2C **(I)**, and 3D **(J)** proteins. ns: $p > 0.05$, $^*p < 0.05$, $^{**}p < 0.01$, $^{***}p < 0.001$, $^{****}p < 0.0001$ by Mann–Whitney test following multiple test correction. The data underlying this figure can be found in S1 Data, pages 11–20.
(TIFF)

**S5 Fig. The fitness effects of single codon deletions across the full CVB3 proteome. (A)** The effects of non-synonymous mutations (ns-mutations) versus deletions across the nonstructural proteins. **(B, C)** Mapping of dMFE in P2 **(B)** and P3 **(C)** derived proteins. The data underlying this figure can be found in S1 Data, page 21.
(TIFF)

**S6 Fig. Correlation between replicates of mutation, site, and deletion MFE in RPE cells. (A)** Correlations matrices for MFE of mutations (mut MFE), their average per site (site MFE), and deletions (dMFE) for independent replicate lines (L1-L3) for the P1, P2, and P3 regions. Of note, for P1, only 2 replicates were used and deletions were not included in the mutagenesis protocol, precluding their analysis.
(TIFF)

**S7 Fig. Analysis of CAR and DAF expression on HeLa-H1 and RPE cells. (A)** The complete membrane is shown for western blots of CAR **(A)** and DAF **(B)** in both cell lines. Arrows indicate bands of the expected size for each protein. A higher molecular weight cross-reactive band is observed for CAR in HeLa-H1 cells. **(C)** Values obtained for the quantification of protein bands. Exposure times of blots used for each protein are indicated.
(TIFF)

**S1 Data. Data underlying Figs 2B–2G, 3A, 3B, 4A–4G, 5B, 5D, S1A, S1B, S3, S4A–S4J, and S5A.**
(XLSX)

**S1 Table. Related to Fig 1.** Primers utilized in the amplification of the synthetic oligonucleotides and the corresponding region in the infectious clone that were used to generate the mutagenized P2 and P3 libraries.
(XLSX)

**S2 Table. Related to Fig 1.** NGS summary statistics for the mutagenized plasmid libraries and the passage 2 virus populations derived in HeLa-H1 cells.
(XLSX)

**S3 Table. Related to Figs 2 and 3.** Site MFE data, predicted ΔΔG, Shannon entropy, and domain annotations for each residue of the CVB3 genome.
(CSV)

**S4 Table. Related to Figs 2, 3, and 4.** Mutation MFE data, predicted ΔΔG, Shannon entropy, and domain annotations for all non-synonymous mutations and deletions across the CVB3 genome.
(CSV)

**S5 Table. Results of the phyDMS analysis for each region of the CVB3 proteome.**
(XLSX)

**S6 Table. Related to Fig 1.** NGS summary statistics for the mutagenized plasmid libraries and the passage 2 virus populations derived in RPE cells.
(XLSX)

**S7 Table. Related to Fig 5.** Site differential selection values for HeLa-H1 and RPE cells.
(CSV)

**S8 Table. Related to Fig 6.** MFE values of predicted druggable pockets.
(XLSX)

**S1 Raw Images. Figure S7 raw, uncropped gels.**
(PDF)

## Acknowledgments

We would like to thank Drs. Rafael Sanjuan and Santiago F. Elena for their helpful suggestions on data interpretation and Drs. Rafael Sanjuan and Olve Peersen for their critical reading of the manuscript. The computations were performed on the HPC cluster Garnatxa at the Institute for Integrative Systems Biology (I2SysBio), a mixed research center formed by the University of Valencia (UV) and the Spanish National Research Council (CSIC). In addition, the authors would like to acknowledge the Principe Felipe Research Center (CIPF) server for Alphafold2 predictions, which was co-financed by the European Union through the Operativa Program of the European Regional Development Fund (ERDF/FEDER) of the Comunitat Valenciana 2014–2020.

## Author Contributions

**Conceptualization:** Beatriz Álvarez-Rodríguez, Ron Geller.

**Data curation:** Ron Geller.

**Formal analysis:** Beatriz Álvarez-Rodríguez, Ron Geller.

**Funding acquisition:** Ron Geller.

**Investigation:** Beatriz Álvarez-Rodríguez, Sebastian Velandia-Álvarez, Ron Geller.

**Methodology:** Beatriz Álvarez-Rodríguez, Sebastian Velandia-Álvarez, Ron Geller.

**Resources:** Ron Geller.

**Software:** Christina Toft.

**Supervision:** Ron Geller.

**Writing – original draft:** Beatriz Álvarez-Rodríguez, Ron Geller.

**Writing – review & editing:** Ron Geller.

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
