## [Editor Report · Decision Letter 0]

8 Apr 2024

Dear Dr Geller, 

Thank you for submitting your manuscript entitled "Mapping the mutational landscape of a viral proteome reveals distinct profiles of mutation tolerability" for consideration as a Research Article by PLOS Biology. I would like to apologize for the delay in coming back to you with an initial decision.

Your manuscript has now been evaluated by the PLOS Biology editorial staff, as well as by an academic editor with relevant expertise, and I am writing to let you know that we would like to send your submission out for external peer review.

Once your full submission is complete, your paper will undergo a series of checks in preparation for peer review. After your manuscript has passed the checks it will be sent out for review. To provide the metadata for your submission, please Login to Editorial Manager (https://www.editorialmanager.com/pbiology) within two working days, i.e. by Apr 10 2024 11:59PM.

Kind regards,

Melissa

Melissa Vazquez-Hernandez, Ph.D.

Associate Editor

PLOS Biology

---

## [Decision Letter · Decision Letter 1]

14 May 2024

Dear Dr Geller,

Thank you for your patience while your manuscript "Mapping the mutational landscape of a viral proteome reveals distinct profiles of mutation tolerability" went through peer-review at PLOS Biology. Your manuscript has now been evaluated by the PLOS Biology editors, an Academic Editor with relevant expertise, and by three independent reviewers.

As you will see in the reports, all reviewers are positive about the relevance of the work, but still some concerns should be addressed prior to publication. Reviewer #1 and #2 require some clarifications. Reviewer #2 would also like to you exclude the possibility of artifacts arising. Reviewer #3 would also like some clarifications regarding Fig 2C. Addressing these and all other concerns of the reviewers is essential for further consideration of your manuscript for publication in PLOS Biology.

**IMPORTANT - SUBMITTING YOUR REVISION**

*Resubmission Checklist*

*Published Peer Review*

*PLOS Data Policy*

*Blot and Gel Data Policy*

Sincerely,

Melissa

Melissa Vazquez Hernandez, Ph.D.

Associate Editor

PLOS Biology

REVIEWERS' COMMENTS

Reviewer #1: 

This is a really excellent deep mutational scanning study of coxsackievirus B3. It is comprehensive, well done, and clearly described. As the authors note, these results are interesting both from the basic perspective of understanding this virus and its proteins, and from the applied perspective of drug design.

I strongly support publication, and only have a few minor comments listed below.

- The authors assembled 250 nt oligos for mutagenesis, which I think (?) is a novel approach. I would have expected this to give a lot of errors in regions flanking the targeted codon, as typically oligo synthesis for very long (250 nt) oligos is quite error prone. However, as best I can tell from Fig S1, this does not seem to be the case. Can the authors provide more details? Perhaps Twist has gotten much more accurate in synthesis?

- Lines 157-159: the text indicates the averaging across replicates was done on the ratios, but was it actually done on the log ratios? If not, maybe it should be averaged on log ratios rather than ratios since the former are what is reported in the paper.

- The correlations between replicates were quite good (Fig S2) for mutations, but are rather poorer for deletions. Why is this?

Reviewer #2: 

This paper from Álvarez-Rodríguez and colleagues uses a synthetic biology-based deep mutational scanning (DMS) approach to compare mutational fitness effects (MFE) of amino acid substitutions and single codon deletions across the entire proteome of coxsackievirus B3. In doing this large-scale comparison, they can quantify how patterns of relative mutational tolerance differ across proteins encoded by the same virus. This is a fascinating question with significant implications for our understanding of viral evolution. 

The authors detail significant differences in mutational tolerance based on specific protein, WT and mutant amino acid, and structural context. Interestingly, they find that the 2A protease is more mutation tolerant than the rest of the viral proteome, while the 3C protease is much less tolerant. They go on to show that MFE profiles are largely similar between viral populations grown in two distinct cell types, with the exception of residues involved in interacting with a viral receptor not present in one of the cell types. Finally, they examine mutational tolerance in 12 "druggable pockets" predicted through structural analyses and detail how these pockets differ in tolerance profiles, with the assumption being that these differences could be used to predict probability of escape from drugs that target these specific pockets. 

Overall, the paper is clearly written, the methods are generally sound (with a couple concerns detailed below) and the conclusions are well supported by the data. Assuming these relatively minor concerns are adequately addressed, I feel this paper will be a significant contribution to the field of viral molecular evolution.

Specific Concerns:

1. While the authors clearly describe their reasonable approach to normalizing fitness effects across the three regions of the viral genome that were experimentally analyzed separately, I think more is needed to further verify the absence of significant artifacts arising from combining data generated from three separate viral libraries/passage experiments. First, it looks like the authors introduced nonsense and synonymous substitutions into the genome. Assuming stop codons will be lethal and silent substitutions generally neutral, can they compare the relative fitness distributions for stop codons and silent substitutions across P1, P2, and P3, before and after linear model-based normalization, to assess how similar they are? 

Also, is it possible to demonstrate how insensitive their findings are to variance in the linear models used to combine the three fragments? Finally, can the authors show the outliers form their comparison of experimental and DMS-derived relative fitness estimates? Is there any reason to believe there could be systematic or biological explanation for divergence in these two measurements of relative fitness that could explain the outliers?

2. In the cell type comparison experiment, the infections in RPE cells were conducted at MOI = 2 (line 694), which is significantly higher than it was in HeLa-H1 and can potentially have a complementary and/or genotype masking effect due to the high probability of coinfection on the population thus confounding the comparison. The authors should either repeat this experiment using equivalent low MOIs in both cell lines.

3. Since RPE cells have a stronger interferon response than HeLa-H1, is it surprising that the authors did not observe differences in mutational tolerance in regions known to be involved in counteracting the IFN response? 

4. Line 212, maybe specify a little more here on the sequence set used rather than just having details in methods. Wasn't clear how many total sequences across how many years were included in final analysis.

5. In figure S1A, please confirm that the number of changes per clone refers to amino acid substitutions. Also, if these are amino acid substitutions, why do so many clones in P1 (where mutations were introduced through PCR) have more than 1 amino acid substitution? I assume that in P2 and P3, multiple mutation containing tiles can get combined into a single haplotype during the PCR process? I may have misunderstood the library construction method, so clarification would be appreciated here.

Reviewer #3: 

Álvarez-Rodríguez et al. perform systematic mutagenesis together with a prior dataset from the group that details the complete mutational landscape of the coxsackievirus B3 proteome. The scale of experiments is impressive, and the authors draw out a number of patterns in the mutagenesis data relative to structural and evolutionary correlates to make sense of these data. The authors also identify variation in mutational tolerance of druggable pockets, illustrating the utility of prospective mutational profiling in selection of potential drug targets with less capacity for evolution of resistance. Overall, the study is well performed and analyzed, and I have just a few minor comments.

Intro highlights the different proteins of CVB3, but then first section of results start discussing "P1", "P2", "P3". It could be quickly included either in the intro or when first mentioning P1-3 how these map to the sets of structural/non-structural proteins themselves.

Most of the structural and evolutionary correlates of mutational tolerance make good sense, but Fig. 2C is difficult to understand how it can show that, for example, mutations to D is maximally deleterious in every instance. It wouldn't be surprising to see some D mutations be deleterious at particular types of amino acid positions, but I wouldn't expect every single mutation to D to be so deleterious at every position tested. (My assumption was this was a box-and-whiskers plots across all measured mutants of different wildtype or mutant classes.) Even mutations like P, G, etc. which are 'unique' amino acids, I wouldn't expect there to be universal detrimental effect at every single position, but especially for a more common amino acid like D, I find it hard to understand that all mutations to D are deleterious. In fact the heatmap in Fig. 2B shows plenty of 'white' scattered through the strip of D mutants, therefore it's not clear why they are absent in the box and whiskers plot. Am I interpreting Fig. 2C correctly? I feel like there must be an error here.

Was any analysis done with respect to synonymous mutations with unexpected deleterious effects that could point to interesting constraints on underlying nucleic acid structures? Though I see in the Methods that synonymous mutations were not routinely included since full NNN mutagenesis was not performed.

Main text could describe what software or method is used to predict druggable pockets, or at least give intuition for what the Schrodinger software is looking for when it predicts pockets.

---

## [Editor Report · Decision Letter 2]

10 Jun 2024

Dear Dr Geller,

Thank you for your patience while we considered your revised manuscript "Mapping the mutational landscape of a viral proteome reveals distinct profiles of mutation tolerability" for publication as a Research Article at PLOS Biology. This revised version of your manuscript has been evaluated by the PLOS Biology editor and the Academic Editor.

Based on our Academic Editor's assessment of your revision, we are likely to accept this manuscript for publication. Please also make sure to address the following data and other policy-related requests.

a) We would like to suggest the following modification to the title:

"Comprehensive analysis of mutational effects across a whole coxsackievirus proteome"

Please supply the numerical values either in the a supplementary file or as a permanent DOI’d deposition for the following figures:

Figure 2BCDEFG, 3AB, 4ABCDEFG, 5BD, S1AB, S3, S4ABCDEFGHIJ, S5A

c) Please cite the location of the data clearly in all relevant main and supplementary Figure legends, e.g. “The data underlying this Figure can be found in S1 Data” or “The data underlying this Figure can be found in https://doi.org/10.5281/zenodo.XXXXX”

d) Please ensure that your Data Statement in the submission system accurately describes where your data can be found and is in final format, as it will be published as written there.

e) Many thanks for providing the underlying code in GitHub. However, because Github depositions can be readily changed or deleted, please make a permanent DOI’d copy (e.g. in Zenodo) and provide this URL in the manuscript and Data Availability Statement.

We expect to receive your revised manuscript within two weeks. 

*Published Peer Review History*

*Press*

Sincerely,

Melissa

Melissa Vazquez Hernandez, Ph.D.

Associate Editor

PLOS Biology

---

## [Editor Report · Decision Letter 3]

13 Jun 2024

Dear Dr Geller,

Thank you for the submission of your revised Research Article "Mapping mutational fitness effects across the coxsackievirus B3 proteome reveals distinct profiles of mutation tolerability" for publication in PLOS Biology. On behalf of my colleagues and the Academic Editor, Jason Ladner, I am pleased to say that we can in principle accept your manuscript for publication, provided you address any remaining formatting and reporting issues. These will be detailed in an email you should receive within 2-3 business days from our colleagues in the journal operations team; no action is required from you until then. Please note that we will not be able to formally accept your manuscript and schedule it for publication until you have completed any requested changes.

PRESS

Sincerely, 

Melissa

Melissa Vazquez Hernandez, Ph.D., Ph.D.

Associate Editor

PLOS Biology
